# Overexpression of *BnaXTH22* Improving Resistance to Aluminum Toxicity in Rapeseed (*Brassica napus* L.)

**DOI:** 10.3390/ijms26125780

**Published:** 2025-06-16

**Authors:** Paolan Yu, Depeng Han, Ming Chen, Lei Yang, Yazhen Li, Tianbao Huang, Wen Xiong, Yewei Cheng, Xiaosan Liu, Changyan Wan, Wei Zheng, Xiaojun Xiao

**Affiliations:** Key Laboratory of Arable Land Improvement and Quality Improvement of Jiangxi Province, Jiangxi Institute of Red Soil and Germplasm Resources, Nanchang 330046, China; 15270904956@163.com (P.Y.); handepeng1113@163.com (D.H.); mingchen8454@126.com (M.C.); jxmhsyl@126.com (L.Y.); liyazhen626@163.com (Y.L.); htb1234@163.com (T.H.); 15179121371@163.com (W.X.); 18407851644@163.com (Y.C.); lxs_0704@163.com (X.L.); 13576952909@163.com (C.W.)

**Keywords:** *Brassica napus*, aluminum toxicity, *BnaXTH22*, overexpressed line

## Abstract

The cell wall, acting as the first line of defense against aluminum (Al) toxicity, is the primary cellular structure that encounters and perceives Al^3+^. Xyloglucan endotransglucosylase/hydrolase (XTH) plays a pivotal role in mediating cell wall remodeling, a critical mechanism for Al toxicity tolerance. In our previous studies, the candidate gene *BnaXTH22* was identified through GWAS and RNA-seq analyses. Under Al toxicity stress, overexpression lines (OEs) exhibited a significant increase in the relative elongation of taproots (9.44–13.32%) and total root length (8.15–12.89%) compared to the wild type (WT). Following Al treatment, OEs displayed reduced MDA content and lower relative electrical conductivity, alongside a significantly higher root activity than WT. Transcriptomic analysis revealed that differentially expressed genes in OE under Al toxicity were predominantly enriched in stress-related biological processes, including phenylpropanoid metabolism, fatty acid biosynthesis, and lignin biosynthesis. These results suggest that *BnaXTH22* overexpression could enhance Al toxicity tolerance in rapeseed, potentially by modulating cell wall synthesis to bolster plant resistance.

## 1. Introduction

Aluminum (Al) is the most abundant metallic element in the Earth’s crust. The exchangeable Al ions (primarily Al(OH)_2_^+^, Al(OH)^2+^, and Al(H_2_O)^3+^) released from silicates or oxides can promote exponential and cause Al toxicity when the soil pH value is below 5.5 [1,2]. Under Al toxicity stress, the first and most important symptom of crops is the inhibition of root elongation, ultimately affecting water and nutrient uptake [3,4,5]. Unfortunately, most of the winter rapeseed areas in the Yangtze River basin of China, which are one of three major rapeseed producing areas in the world, grow in the acidic red soil areas with pH 5.04~5.37. Additionally, approximately 40% of the world’s potential cultivable land exhibits a pH below 5.5 [2,5,6,7]. In these acidic conditions, exchangeable Al ions precipitate in the soil, creating Al toxicity stress that severely restricts the growth and yield of rapeseed [5,8]. As a result, Al toxicity has become a major limiting factor for crop productivity in acidic soils worldwide.

Plant tolerance to Al is a complex trait, which is controlled by multiple genes and pathways [9,10,11,12,13]. Xyloglucan endotransglucosylase/hydrolase (XTH) proteins, encoded by a polygenic family and belonging to the glycoside hydrolase family GH16, play a crucial role in mediating cell wall remodeling [13,14]. Different members of the XTH family genes exhibit distinct functions in regulating Al toxicity tolerance. For instance, *XTH31* has been shown to participate in the regulation of Al tolerance in *Arabidopsis* roots. Compared to the wild type, *xth31* mutants accumulate less Al in their roots and root cell walls, demonstrating enhanced Al tolerance [15]. Similarly, *XTH17* has been identified to function analogously to *XTH31* and is also involved in the response to Al toxicity [16,17]. Overexpression of *AtXTH32* in *Arabidopsis* significantly inhibits root growth and triggers typical markers of programmed cell death (PCD), whereas suppression of *xth32* results in largely opposite effects [18]. In contrast, *ZmXTH* has been found to enhance Al tolerance in transgenic *Arabidopsis* by reducing Al accumulation in their roots and cell walls [19].

In previous studies, *BnaA10g11500D* was identified through multi-omics analysis as potentially related to Al tolerance, and this gene may play an important role in the response of rapeseed to Al toxicity stress [20]. As the homologous gene of *BnaA10g11500D*, *AtXTH22* encodes the TCH4 enzyme, the substrate of the TCH4 enzyme is xyloglucan, which is closely related to cell wall stretching [21]. *BnaA10g11500D* is named *BnaXTH22*. The Al toxicity tolerance function of *BnaXTH22* has not been reported in rapeseed. Combined with the expression level and functional annotation of *BnaXTH22*, overexpression genetic transformation was conducted in rapeseed for verification. Then, the phenotype and physiological response of overexpressed lines (OEs) were identified and measured, and transcriptome analysis was carried out. This study significantly expands our understanding of the physiological and molecular mechanisms underlying the Al toxicity tolerance of *BnaXTH22* overexpressing lines.

## 2. Result

### 2.1. Analysis of the Expression Pattern of the BnaXTH22 Under Al Toxicity Stress

*BnaXTH22* was differentially expressed at both 6 h vs. 0 h and 24 h vs. 0 h in both R178 (aluminum tolerant line; ATL) and S169 (aluminum sensitive line; ASL). To validate its expression under Al toxicity stress, we performed qRT-PCR analysis. These results demonstrated expression patterns consistent with the RNA-seq data, confirming the reliability of the selected differentially expressed genes (DEGs) (Figure 1). These findings indicate that *BnaXTH22* likely played an important functional role in rapeseed response to Al toxicity stress.

### 2.2. Generation of Transgenic Plants and Molecular Identification

The recombinant plasmid pCAMBIA1301-BnaXTH22 (Figure 2A) was successfully constructed and transformed into wild-type rapeseed (Westar, non-transgenic control, WT). Eight independent transgenic lines were established, with successful transformation confirmed through T_0_ generation screening (Figure 2B). Three randomly selected T_3_ transgenic lines (OE-2, OE-4, and OE-6) were subjected to *BnaXTH22* expression analysis. Compared to WT, the expression level of these overexpression lines (OEs) exhibited 2.47- to 6.64-fold higher *BnaXTH22* expression levels in both leaves and roots (Figure 2C), confirming successful transgene overexpression. Three lines (OE-2, OE-4, and OE-6) were consequently chosen for further experimental investigations.

### 2.3. Phenotype Characterization of Overexpressing BnaXTH22

To assess phenotypic and physiological responses to 60 μM AlCl_3_ treatment, we analyzed OE-2, OE-4, OE-6, and WT. The OE showed no significant difference in the taproot length and total root length compared with the WT under normal (non-Al stress) conditions. The relative elongation of taproots (RET) in OE lines (0.649, 0.672, and 0.657 for OE-2, OE-4, and OE-6, respectively) showed significant increases of 9.44%, 13.32%, and 10.79%, respectively, compared to WT (0.593) (Figure 3A,B). While no significant differences existed among OE lines, all demonstrated markedly greater RET than WT. Similarly, the relative total root length (RTRL) measurements revealed values of 0.741, 0.730, and 0.762 for OE-2, OE-4, and OE-6, respectively, corresponding to 9.78%, 8.15%, and 12.89%, respectively, increases over WT (0.675) (Figure 3C). Although RTRL did not differ significantly among OE lines, all transgenic plants exhibited significantly greater values than WT, mirroring the RET pattern. Notably, RTRL exceeded RET in both WT and OE, suggesting that Al toxicity exerts a more pronounced inhibitory effect on taproot growth than on lateral roots.

### 2.4. MDA, REC, and RA of OEs Response to Al Toxicity Stress

Under control conditions (0 h treatment), no significant differences were observed in any physiological indices between OEs and WT. However, following exposure to 60 μM AlCl_3_ stress, while the OEs showed no significant variation among themselves, they collectively demonstrated markedly enhanced stress tolerance compared to WT, as evidenced by all of the physiological parameters measured (Figure 4).

The content of MDA increased when the plants were exposed to Al toxicity stress, and there was a significant difference in the content of MDA between WT and OEs after 7 days of treatment (Figure 4A). The relative electrical conductivity (REC) increased when the plants were exposed to Al toxicity stress. The OEs exhibited lower levels of stress-induced ion leakage compared to WT and there was a significant difference in the relative electrical conductivity between WT and OEs after 7 days of treatment (Figure 4B). The root activity (RA) was opposite to the content of MDA and the relative electrical conductivity. The root activity reduced when the plants were exposed to Al toxicity stress. The OEs exhibited higher levels of root activity compared to WT between WT and OEs after 24 h and 7 days of treatment (Figure 4C).

### 2.5. Transcriptome Analysis of Overexpressing BnaXTH22

The RET of OE-2 and WT was 0.621 ± 0.034 and 0.533 ± 0.042, respectively, under 60 μM AlCl_3_ for 24 h. The RET of OE-2 increased significantly by 15.23% compared to WT. To obtain greater insight into the underlying molecular mechanism modulating Al tolerance and stress-responsive genes by *BnaXTH22* overexpression, RNA-seq was conducted with OE-2 and WT as the experimental materials. More than 525.2 million clean reads from 12 libraries of the two genotypes were generated and mapped to the reference genome (Table 1).

### 2.6. Al Toxicity Response Related Genes with BnaXTH22 Overexpression

To determine the genes correlated with *BnaXTH22* overexpression, the DEGs between the OE-2 and WT for 0 h and 24 h were identified. After screening by |log2 fold change| > 1.0 and *p* value < 0.05, a total of 4877 DEGs between the OE-2 and WT, of which 2197 DEGs were up-regulated and 2680 DEGs were down-regulated. In 0 h, a total of 3827 DEGs between the OE-2 and WT, of which 1737 DEGs showed up-regulation and 2090 DEGs showed down-regulation. Whereas 565 DEGs showed up-regulation and 745 DEGs showed down-regulation at 24 h (Figure 5A). Interestingly, 260 common DEGs were identified between the OE-2 and WT at both 0 h and 24 h (Figure 5B).

The GO enrichment analysis of 3827 DEGs with WT and OE-2 at 0 h showed that these DEGs participated in a variety of biological processes, including DNA-dependent DNA replication, DNA replication, water transport, liquid transport, etc. (Figure 5C). Under the condition of Al toxicity treatment for 24 h, it mainly participated in the phenylpropanoid biosynthetic process, fatty acid biosynthetic process, suberin biosynthetic process, phenylpropanoid biosynthetic process, phenylpropanoid metabolic process, and other adverse biological processes (Figure 5D).

The pathway enrichment analysis of WT and OE-2 at 0 h showed that these DEGs were significantly enriched in arginine and proline metabolism, arginine biosynthesis, and purine metabolism (Figure 5E). Under the condition of Al poisoning for 24 h, alanine, aspartate, and glutamate metabolism; arginine biosynthesis; arginine and proline metabolism; and pyruvate metabolism were significantly enriched (Figure 5F).

## 3. Discussion

Plant cell walls are composed of various molecules and have many important physiological functions. One of the factors responsible for the plasticity of the cell wall is the XTH family, which cleaves and reconnects xyloglucan molecules. XTHs are not necessary for cell wall loosening during plant cell expansion, but play a critical role under specific biological or abiotic stresses [22]. Compared with the wild type, *xth31* mutants accumulate less Al in roots and root cell walls, and show stronger Al tolerance in *Arabidopsis* [15]. *XTH17* and *AhXTH32* were found to be similar to *XTH31* and are involved in Al toxicity response [16,17,18]. Meanwhile, *ZmXTH* could confer the Al tolerance of transgenic *Arabidopsis* by reducing the Al accumulation in their roots and cell walls [19]. Combined with the expression level and functional annotation of *BnaXTH22*, overexpression genetic transformation was conducted in rapeseed for verification.

Root growth and elongation are the synergistic result of root cell division and cell elongation. The earliest morphological changes of crops induced by Al are root and shoot growth inhibition. Root cap cells, meristem cells, tanycytes, and root hair cells are the most severely affected parts [23,24]. Al interferes with cell division in root tips and lateral roots by inhibiting the production and transport of cytokinin [23,25]. Al toxicity stress affects the transport of auxin between cells, thus inhibiting root growth [26,27,28]. The Al tolerance of crops (including rape) is often expressed or identified by root-related traits such as taproot length and taproot relative productivity at the seedling stage, and based on this, resource identification, gene mining, and functional verification are carried out [29,30,31]. In this study, it was found that the RTE of rape seedlings treated with Al toxicity was significantly reduced, which was consistent with the conclusion of the previous study that the RTE of rape was gradually reduced under Al toxicity stress [32], and a similar pattern was found in *Arabidopsis*, rice, lettuce, and other crops [33,34,35]. However, in this study, RTE and RTRL of OEs treated with 60µM AlCl_3_ were significantly higher than those of WT, indicating that the resistance of OEs to Al toxicity was significantly better than that of WT, indicating that overexpression of *BnaXTH22* improved the Al toxicity resistance of rape.

MDA is the end-product of membrane lipid peroxidation and can be used to characterize the degree of lipid peroxidation of the plant cell membrane. Relative electrical conductivity is a basic index to reflect the permeability of the plant cell membrane, and the root activity level is an important index to reflect the root growth status. Under stress such as Al toxicity, the MDA content increased, relative conductivity increased, and root activity decreased [35,36,37,38]. In this study, the MDA content of WT and OEs increased, the relative conductivity increased, and the root activity decreased under Al toxicity stress. Compared with WT, the MDA content after 7 days of OEs treatment was lower, and the increase in MDA content was smaller than that after 0 h treatment. The REC of OEs after 7 days increased less than that of 0 h, and the REC of OEs was lower or significantly lower than that of WT. The root activity of WT was lower or significantly lower than that of OEs after 7 days of treatment. In conclusion, the overexpression of *BnaXTH22* increased the resistance of rapeseed to Al toxicity.

RNA-seq was used to identify differentially expressed genes in the roots and leaves of rapeseed seedlings under Al toxicity and drought stress, and the molecular functions and metabolic pathways of related differentially expressed genes were clarified through GO and KEGG enrichment analysis [39,40,41]. The overexpressed transcriptome can be used to analyze the regulatory mechanism of target gene overexpression under the same genetic background. Transcriptome analysis of *PpSAUR73*-overexpressing *Arabidopsis thaliana* was performed to compare differentially expressed genes between overexpressed plants and wild-type plants, and significant enrichment analysis of GO function and KEGG pathway was performed to determine that *PpSAUR73* can regulate growth of *Arabidopsis thaliana* and participate in various hormone signal transductions [42]. Transcriptome sequencing was performed on plants overexpressing *DgLsL* in different treatments, and the enrichment analysis of GO and KEGG pathways confirmed previous studies on axillary bud germination and growth, revealing the important role of genes involved in plant hormone biosynthesis and signal transduction [43]. In this study, GO enrichment analysis showed that the regulation of abiotic stress such as plant Al toxicity involved multiple functional groups. One of them affects plant type secondary cell wall biosynthesis [44]. Some NAC and MYB transcription factors are involved in the biosynthesis of lignin, cellulose, and xylan in cell wall [45,46,47,48], and irregular xylem proteins are also involved in xylan biosynthesis [49]. Arabinogalactan, a bunch-like protein, has been shown to play an important role in plant development and environmental adaptation [50]. In this study, it was found that DEGs are mainly involved in stress biological processes such as phenylpropanoid metabolism, fatty acid biosynthesis, lignin biosynthesis, and phenylC biosynthesis under Al toxicity treatment, which was conducive to enhancing Al toxicity tolerance of plants overexpressing *BnaXTH22*.

## 4. Materials and Methods

### 4.1. Validation of Gene by qRT-PCR

ATL and ASL were used as materials for cultivation, treatment, and qRT-PCR by referring to our previous studies [20]. Here, we chose the candidate gene *BnaXTH22*, which was simultaneously detected between 6 h vs. 0 h and 24 h vs. 0 h in ATL and ASL, respectively. The primer sequences used for qRT-PCR can be found in Table 2. The relative expression levels were determined using the 2^−ΔΔCt^ method based on the normalization to the reference genes *ACT7*. Three technical replicates were performed for each DEG and reference gene.

### 4.2. Generation of Transgenic Westar Plants

Based on the total RNA of the root after 24 h of ATL Al stress, we used reverse transcription into cDNA using reverse transcription kit (Beijing Baori Medical Technology Co., Ltd., Beijing, China). Using cDNA as a template, *BnaXTH22* CDS of 645 bp was obtained by PCR amplification with specific forward primers and reverse primers. The nucleotide sequence of the forward primer is 5′-ATGCAGATGAAACTCGTCC-3′. The nucleotide sequence of the reverse primer is 5′-CTATGCAGCAAAGCACTCTT-3′. Then, *BnaXTH22* CDS is linked to the overexpression vector pCAMBIA1301. Genetic transformation into wild-type rape Westar was performed by Wuhan Biorun Biosciences Co., Ltd. (Wuhan, China). After harvesting positive OEs of T0 generation, positive plants of T1 generation and T2 generation were obtained by continuous self-fertilization. The positive test plants of PCR product 683 bp were selected with a forward primer 5′-TGACGTAAGGGATGACGCAC-3′ and reverse primer 5′-TGGGAGTTCCATCGACTGTG-3′. The cDNA of transgenic rapeseed plants and wild rapeseed plants were used as templates for qRT-PCR verification. The internal reference gene is *ACT7*. Finally, the overexpressed positive seeds of the T3 generation were used for further experiments.

### 4.3. Morphological and Physiological Parameter Under Al Stress

Phenotypic identification and physiological response processing time was 7 days by 60 μmol·L^−1^ AlCl_3_. The hydroponic method and index calculation is the same as our previous studies [20]. Physiological parameters of WT and OEs rapeseed root were measured, including MDA content, relative electrical conductivity, and root activity. These physiological indicators are carried out according to the kit instructions (Suzhou Grace Biotechnology Co., Ltd., Suzhou, China).

### 4.4. RNA-Seq Under Al Tolerance and Data Analysis

To obtain greater insight into the underlying molecular mechanism modulating Al tolerance and stress-responsive genes by *BnaXTH22* overexpression, OE-2 and WT were treated with 60 µmol·L^−1^ AlCl_3_ for 0 h and 24 h, respectively. Then, the roots were quickly frozen in liquid nitrogen. To detect the differentially expressed genes in different treatments we used RNA-seq. Sequencing service was provided by Wekemo Tech Group Co., Ltd., Shenzhen, China. The data and analysis methods were obtained by referring to our previous studies [20]. The RET of the OE and WT was measured at 24 h.

## 5. Conclusions

Compared to the wild type (WT), overexpressing *BnaXTH22* exhibited significantly greater relative elongation of taproots and total root length, reduced MDA accumulation and relative electrical conductivity, markedly higher root activity, and enhanced tolerance to Al toxicity. Transcriptomic analysis revealed that *BnaXTH22* overexpression upregulates stress-related biological processes, including phenylpropanoid metabolism, fatty acid biosynthesis, lignin biosynthesis, and phenylpropanoid biosynthesis, thereby improving Al toxicity resistance.

## Figures and Tables

**Figure 1 ijms-26-05780-f001:**
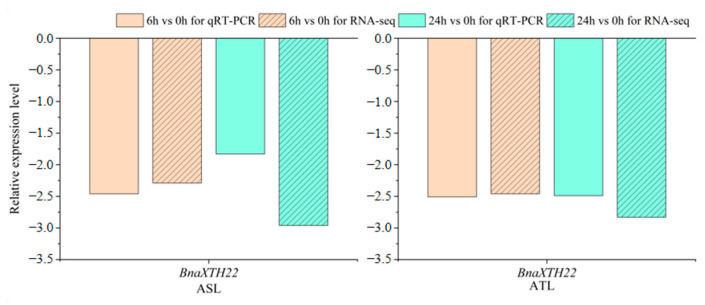
Expression of *BnaXTH22* in rape root by qRT-PCR and RNA-seq.

**Figure 2 ijms-26-05780-f002:**
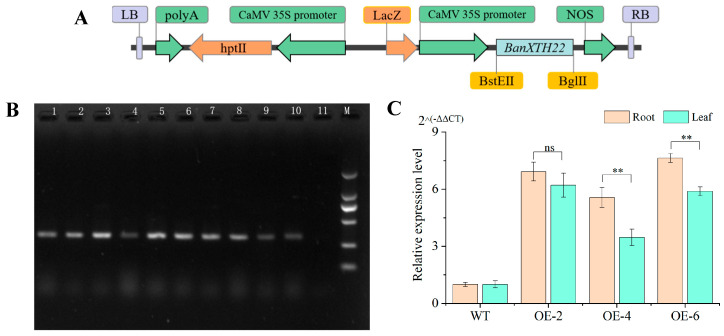
Screening and quantitative analysis of transgenic positive plants of *BnaXTH22*. (**A**) The vector map of pCAMBIA1301-BnaXTH22. (**B**) PCR detection of *BnaXTH22* in transgenic Westar. Lane 1 to lane 9 represent 9 genetically transformed plants. Lane 10 is the recombinant plasmid pCAMBIA1301-BnaXTH22. Lane 11 is the acceptor material Westar. Lane M is the 2kb marker. (**C**) The relative expression level of *BnaXTH22* of overexpression plant. Significance analysis was performed using *t*-test (ns *p* > 0.05, ** *p* < 0.01).

**Figure 3 ijms-26-05780-f003:**
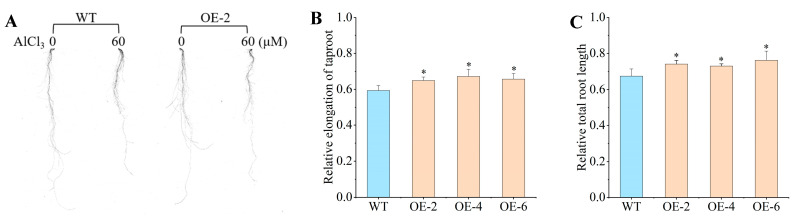
The phenotype traits in the WT and OEs under Al toxicity stress. (**A**) The WT and OE-2 were under 60 µM AlCl_3_ treatment. (**B**) The RET statistics of WT and OEs under 60 µM AlCl_3_ treatment. (**C**) The RTRL statistics of WT and OEs under 60 µM AlCl_3_ treatment. Significance analysis was performed between WT and OE using *t*-test (* *p* < 0.05).

**Figure 4 ijms-26-05780-f004:**
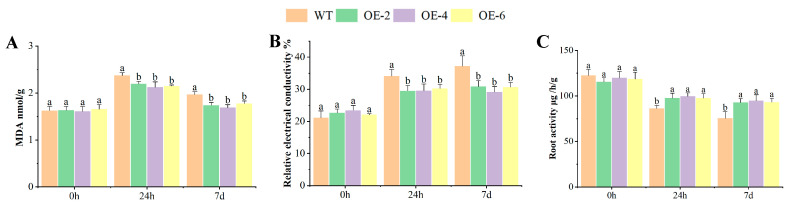
The physiological response in the OEs and WT under Al toxicity stress. (**A**) The content of MDA in rapeseed root. (**B**) The relative electrical conductivity in rapeseed root. (**C**) The root activity in rapeseed root. Significance analysis was performed between WT and OE using *t*-test under processing at the same time (different small letters; *p* < 0.05).

**Figure 5 ijms-26-05780-f005:**
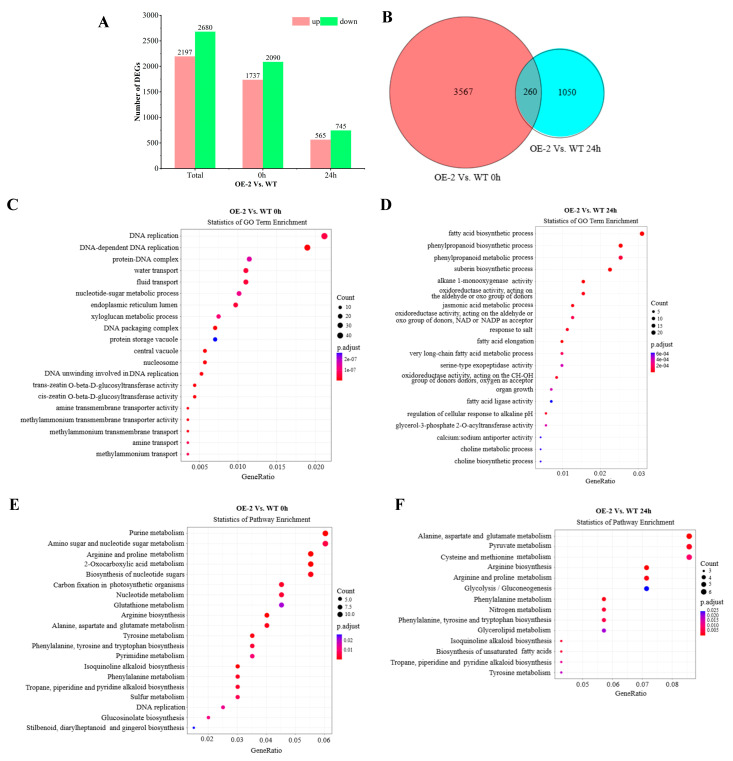
Analysis of the DEGs at 24 h and 0 h under Al toxicity treatment of OE-2 and WT. (**A**) The distribution map of DEGs. (**B**) Venn diagram of DEGs. (**C**,**D**) GO enrichment analysis. (**E**,**F**) Pathway enrichment analysis.

**Table 1 ijms-26-05780-t001:** The data of RNA-seq samples.

Sample	Clean Reads	Clean Bases	Proportion of Q30	Mapped Ratio	GC Content
WT 0 h-1	42,143,362	6,044,721,779	0.934	0.910	0.467
WT 0 h-2	43,114,336	6,185,668,673	0.934	0.906	0.466
WT 0 h-3	41,970,324	5,993,085,324	0.927	0.905	0.466
WT 24 h-1	42,655,310	6,122,028,343	0.931	0.913	0.461
WT 24 h-2	40,637,354	5,813,384,648	0.935	0.904	0.465
WT 24 h-3	44,551,704	6,365,419,321	0.933	0.903	0.466
Total	255,072,390	36,524,308,088			
OE-2 0 h-1	42,527,036	6,069,309,557	0.938	0.904	0.468
OE-2 0 h-2	42,746,136	6,093,607,070	0.929	0.904	0.467
OE-2 0 h-3	42,817,394	6,145,304,675	0.933	0.905	0.467
OE-2 24 h-1	54,965,910	7,885,013,726	0.935	0.917	0.462
OE-2 24 h-2	44,846,140	6,395,780,703	0.937	0.899	0.471
OE-2 24 h-3	42,181,708	6,004,578,510	0.920	0.902	0.468
Total	270,084,324	38,593,594,241			

**Table 2 ijms-26-05780-t002:** The primer sequences of qRT-PCR.

Gene	Forward Primer (5′-3′)	Reverse Primer (5′-3′)	Size/bp
*BnaXTH22*	CACGAGAGGTGGTTTGGTCA	GAGCCGTAGAGTCAAGCTCC	173
*ACT7*	CCTCTCAACCCGAAAGCCAA	CATCACCAGAGTCGAGCACA	148

## Data Availability

The data presented in this study are available in this article.

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
