# Peer review of "Overexpression of BnaXTH22 Improving Resistance to Aluminum Toxicity in Rapeseed (Brassica napus L.)"

_ijms, 2025, doi:10.3390/ijms26125780_

Round 1
Reviewer 1 Report
Comments and Suggestions for Authors
The manuscript presents a standard functional analysis of the BnaXTH22 gene in rapeseed, focusing on its role in improving aluminum (Al) toxicity resistance through overexpression studies. The research is methodologically sound, and the results support the conclusion that BnaXTH22 contributes to Al stress tolerance by affecting root growth and related physiological traits. However, the study lacks novelty, as the approach and findings are relatively predictable and similar to many prior overexpression studies. The manuscript would benefit from improved language clarity and more in-depth discussion of the molecular mechanisms. Nevertheless, the work may be of interest to readers in the field of plant stress physiology, particularly those working on cell wall-related responses to Al toxicity..
Major Comments:
- The study follows a conventional overexpression workflow—gene selection, transgenic validation, phenotype assays, and transcriptomics. Similar approaches and conclusions have been reported in studies of other XTH family genes. The authors should emphasize what is new or unique about BnaXTH22 compared to other known XTHs, particularly in rapeseed.
- In lines 57–60, the authors state: “In previous studies, our research group identified the gene BnaA10g11500D through multi-omics analysis as potentially related to Al tolerance, and we discovered that BnaA10g11500D may played an important role in the response of rapeseed to Al toxicity stress.” However, it is known that the previous study identified four candidate genes. The rationale for selecting BnaA10g11500D specifically for further functional validation is not clearly explained in this manuscript. Was the selection based on prior functional verification of its homologs in other species (e.g., Arabidopsis or maize)? A clearer justification for prioritizing this gene over the others should be provided to strengthen the logic of the study design.
- BnaXTH22 encodes a xyloglucan endotransglucosylase/hydrolase (XTH), which is generally associated with cell wall remodeling. However, the transcriptome analysis revealed 4877 differentially expressed genes (DEGs) between overexpression lines and the wild type under Al stress conditions. This is a surprisingly large number of DEGs for a cell wall-associated gene, and the manuscript does not provide a clear explanation for this broad transcriptomic impact. The authors should address why BnaXTH22 overexpression leads to such widespread changes—are these likely to be direct effects, or secondary consequences due to altered root growth or cell wall structure?
Moreover, although the RNA-seq data are presented, the study lacks a deeper mechanistic explanation of how BnaXTH22 modulates downstream pathways. The discussion only briefly references enriched KEGG categories (e.g., phenylpropanoid biosynthesis, oxidative stress), but does not connect these findings to known functions of XTH genes or propose plausible signaling intermediates. While overinterpretation should be avoided, the authors are encouraged to provide a more grounded and biologically meaningful discussion of how BnaXTH22 might interface with Al stress signaling networks based on existing literature.
Minor Comments:
- Some figure legends are overly brief and lack sufficient detail to allow the reader to interpret the figures independently. For instance, in Figure 2C, the asterisk (*) is not explained in the legend—its meaning (e.g., p-value thresholds) should be clearly defined. Similarly, in Figure 4, the use of lowercase letters to denote statistical differences is not described in the legend. All statistical symbols or annotations used in figures should be explicitly explained. Additionally, Figure 1 presents QPCR data but does not show standard error (SE) or standard deviation (SD).
- is inconsistency in the capitalization of words in reference titles—for instance, words in the titles from Lines 392–393 are capitalized, while those in Lines 394–395 are not. These inconsistencies indicate that the reference formatting requires thorough revision. Please carefully check all references and revise them according to the journal’s formatting requirements.
- The manuscript does not report whether the BnaXTH22 overexpression lines exhibit any phenotypic changes under normal (non-Al stress) conditions. Including this comparison would help determine if the observed effects are specific to aluminum stress or also present under standard growth conditions.
The overall quality of the English language in this article is generally well-written, yet there are certain aspects that necessitate refinement.
- Line 59, “may played an important role” → “may play an important role”.
- Line 226, “Transcriptome analysis of overexpression of PpSAUR73 in Arabidopsis” → “Transcriptome analysis of PpSAUR73-overexpressing Arabidopsis”.
- Line 62, “BnaA10g11500D was named BnaXTH22” → “BnaA10g11500D is named BnaXTH22”.
Please meticulously review the entire text for the accuracy of English tenses and the fluidity of the language and proceed with the necessary revisions.
Author Response
Dear Editor and Reviewers:
We very appreciate for editor’s efforts and are grateful for receiving comments from reviewers. We have revised the manuscript in accordance with the comments of editor and reviewers, and carefully proof-read the manuscript to minimize the possible presented errors. We responded to the questions raised by the reviewers one by one. We have made revisions and retained the traces of the revisions. Here below is our description on revision according to the reviewers’ comments.
Reviewer Comments:
Comments and Suggestions for Authors
The manuscript presents a standard functional analysis of the BnaXTH22 gene in rapeseed, focusing on its role in improving aluminum (Al) toxicity resistance through overexpression studies. The research is methodologically sound, and the results support the conclusion that BnaXTH22 contributes to Al stress tolerance by affecting root growth and related physiological traits. However, the study lacks novelty, as the approach and findings are relatively predictable and similar to many prior overexpression studies. The manuscript would benefit from improved language clarity and more in-depth discussion of the molecular mechanisms. Nevertheless, the work may be of interest to readers in the field of plant stress physiology, particularly those working on cell wall-related responses to Al toxicity
Major Comments:
- The study follows a conventional overexpression workflow—gene selection, transgenic validation, phenotype assays, and transcriptomics. Similar approaches and conclusions have been reported in studies of other XTH family genes. The authors should emphasize what is new or unique about BnaXTH22compared to other known XTHs, particularly in rapeseed.
R: Thank you very much for your professional suggestions. We have added the following description: The Al toxicity tolerance function of BnaXTH22 has not been reported in rapeseed.
- In lines 57–60, the authors state: “In previous studies, our research group identified the gene BnaA10g11500Dthrough multi-omics analysis as potentially related to Al tolerance, and we discovered that BnaA10g11500D may played an important role in the response of rapeseed to Al toxicity stress.” However, it is known that the previous study identified four candidate genes. The rationale for selecting BnaA10g11500D specifically for further functional validation is not clearly explained in this manuscript. Was the selection based on prior functional verification of its homologs in other species (e.g., Arabidopsis or maize)? A clearer justification for prioritizing this gene over the others should be provided to strengthen the logic of the study design.
R: It is known that the previous study identified four candidate genes. The BnaA03g30320D was has been verified by us to be related to aluminium poisoning, but the article has not been published yet. BnaC03g38360D and BnaC06g30030D temporarily not seen resistance related research reports, we are trying to study, but did not see the deterministic result. In this study, selecting BnaA10g11500D was based on prior functional verification of its homologs in other species ( in lines 45–57)
- BnaXTH22 encodes a xyloglucan endotransglucosylase/hydrolase (XTH), which is generally associated with cell wall remodeling. However, the transcriptome analysis revealed 4877 differentially expressed genes (DEGs) between overexpression lines and the wild type under Al stress conditions. This is a surprisingly large number of DEGs for a cell wall-associated gene, and the manuscript does not provide a clear explanation for this broad transcriptomic impact. The authors should address why BnaXTH22overexpression leads to such widespread changes—are these likely to be direct effects, or secondary consequences due to altered root growth or cell wall structure?
R: Gene profiling of OE versus WT lines revealed specific genetic reprogramming under Al stress, with enrichment in lignin, phenylpropanoid and fatty acid biosynthetic pathways - essential for cell wall reinforcement and stress adaptation. BnaXTH22 overexpression leads to such widespread changes—are these likely to be direct effects, or secondary consequences due to altered root growth or cell wall structure, also be a combined direct and indirect influence. The cause needs to be explored and further systematic research is required.
Minor Comments:
- Some figure legends are overly brief and lack sufficient detail to allow the reader to interpret the figures independently. For instance, in Figure 2C, the asterisk (*) is not explained in the legend—its meaning (e.g., p-value thresholds) should be clearly defined. Similarly, in Figure 4, the use of lowercase letters to denote statistical differences is not described in the legend. All statistical symbols or annotations used in figures should be explicitly explained. Additionally, Figure 1 presents QPCR data but does not show standard error (SE) or standard deviation (SD).
R: Thank you very much for your professional suggestions. The issues regarding the interpretation of the legends pointed out by the review experts have been modified as required, and other legend checks and corresponding modifications have been carried out. Figure 1 presents the QPCR data. Since it needs to be unified with the transcriptome data, the standard error (SE) or standard deviation (SD) is not shown.
- is inconsistency in the capitalization of words in reference titles—for instance, words in the titles from Lines 392–393 are capitalized, while those in Lines 394–395 are not. These inconsistencies indicate that the reference formatting requires thorough revision. Please carefully check all references and revise them according to the journal’s formatting requirements.
R: Thank you very much for your professional suggestions. The issues pointed out by the review experts have been revised as required by the journal, and all references have been verified.
- The manuscript does not report whether the BnaXTH22overexpression lines exhibit any phenotypic changes under normal (non-Al stress) conditions. Including this comparison would help determine if the observed effects are specific to aluminum stress or also present under standard growth conditions.
R: Thank you very much for your professional suggestions. At present, there is no significant difference in the root phenotypic traits at the seedling stage under normal growth conditions. We add relevant descriptions when interpreting Figure 3, but other traits, especially we will conduct further systematic research on various traits such as the flowering period and the maturity period.
Comments on the Quality of English Language
The overall quality of the English language in this article is generally well-written, yet there are certain aspects that necessitate refinement.
- Line 59, “may played an important role” → “may play an important role”.
- Line 226, “Transcriptome analysis of overexpression ofPpSAUR73 in Arabidopsis” → “Transcriptome analysis of PpSAUR73-overexpressing Arabidopsis”.
- Line 62, “BnaA10g11500Dwas named BnaXTH22” → “BnaA10g11500D is named BnaXTH22”.
- R: Thank you very much for your professional suggestions. The review experts pointed out that all the above three opinions have been modified accordingly, andmeticulously review the entire text for the accuracy of English tenses and the fluidity of the language and proceed with the necessary revisions.
Reviewer 2 Report
Comments and Suggestions for Authors
The title of the article is relevant and encourages readers in the field to read it. The material and method of work are well described, but there could be more statistical processing of the data obtained. The experimental design is well done and emphasizes gene identification by integrating GWAS and transcriptomics, supporting the biological relevance of BnaXTH22. At the same time, gene overexpression and phenotype analysis correlated with transcriptomic data add an essential layer of biological confirmation, leading to functional genetic validation. The link between gene expression, cell wall alterations and Al resistance is supported by multiple lines of evidence (phenotypic, biochemical, transcriptomic), thus the physiological context is well correlated with the molecular one. The research would be good to include lines with suppressed expression (RNAi/CRISPR) of BnaXTH22, which could have bidirectionally confirmed the role of the gene in Al tolerance. It is not clear whether overexpression of BnaXTH22 also affects other important traits such as overall growth, productivity/yield, etc., which may limit the applicability of the results in agricultural practice. Since the enzymatic activity of TCH4/XTH22 in OE lines is not directly measured, this aspect leads to the existence of a missing link between gene expression and biochemical function. The research results are relevant and tables and figures are described in the text. Through this study, BnaXTH22 (BnaA10g11500D) gene was identified as a relevant gene for the response to Al by a combination of GWAS and RNA-seq analysis. This type of analysis is based on a robust strategy that correlates gene diversity with differential expression under stress. The choice of this gene was motivated by its functional role (involvement in cell wall synthesis and remodeling through TCH4/xyloglucan endotransglucosylase activity). Gene profiling of OE versus WT lines revealed specific genetic reprogramming under Al stress, with enrichment in lignin, phenylpropanoid and fatty acid biosynthetic pathways - essential for cell wall reinforcement and stress adaptation. The bibliographical references are recent (most of them from the last 5 years) and relevant to the research field. The authors demonstrate good documentation with 50 bibliographical titles.
Author Response
Dear Editor and Reviewers:
We very appreciate for editor’s efforts and are grateful for receiving comments from reviewers. We have revised the manuscript in accordance with the comments of editor and reviewers, and carefully proof-read the manuscript to minimize the possible presented errors. We responded to the questions raised by the reviewers one by one. We have made revisions and retained the traces of the revisions. Here below is our description on revision according to the reviewers’ comments.
Comments and Suggestions for Authors
The title of the article is relevant and encourages readers in the field to read it. The material and method of work are well described, but there could be more statistical processing of the data obtained. The experimental design is well done and emphasizes gene identification by integrating GWAS and transcriptomics, supporting the biological relevance of BnaXTH22. At the same time, gene overexpression and phenotype analysis correlated with transcriptomic data add an essential layer of biological confirmation, leading to functional genetic validation. The link between gene expression, cell wall alterations and Al resistance is supported by multiple lines of evidence (phenotypic, biochemical, transcriptomic), thus the physiological context is well correlated with the molecular one. The research would be good to include lines with suppressed expression (RNAi/CRISPR) of BnaXTH22, which could have bidirectionally confirmed the role of the gene in Al tolerance. It is not clear whether overexpression of BnaXTH22 also affects other important traits such as overall growth, productivity/yield, etc., which may limit the applicability of the results in agricultural practice. Since the enzymatic activity of TCH4/XTH22 in OE lines is not directly measured, this aspect leads to the existence of a missing link between gene expression and biochemical function. The research results are relevant and tables and figures are described in the text. Through this study, BnaXTH22 (BnaA10g11500D) gene was identified as a relevant gene for the response to Al by a combination of GWAS and RNA-seq analysis. This type of analysis is based on a robust strategy that correlates gene diversity with differential expression under stress. The choice of this gene was motivated by its functional role (involvement in cell wall synthesis and remodeling through TCH4/xyloglucan endotransglucosylase activity). Gene profiling of OE versus WT lines revealed specific genetic reprogramming under Al stress, with enrichment in lignin, phenylpropanoid and fatty acid biosynthetic pathways - essential for cell wall reinforcement and stress adaptation. The bibliographical references are recent (most of them from the last 5 years) and relevant to the research field. The authors demonstrate good documentation with 50 bibliographical titles.
R: Thank you for the recognition of the reviewer. For evaluation expert advice we main answer is as follows.
- This article only published gene expression was studied, because we tried RNAi and CRISPR without success.
- This paper only studied the root system of rapeseed seedlings, which is the most sensitive to aluminum toxicity response. Later, studies on treatment systems with different aluminum concentrations at different critical periods (flowering period, maturity period) will be carried out to clarify the possibility of their production and application.
- As for TCH4/xyloglucan endotransglucosylase activity, we have not mastered the appropriate method to accurately determine this index.Although this does not affect our judgment of gene function. It is a pity that the evidence is lacking.
Round 2
Reviewer 1 Report
Comments and Suggestions for Authors
The authors have addressed my previous concerns appropriately, and the manuscript has significantly improved in clarity and scientific rigor. I appreciate their efforts in revising the work. I have no further major concerns, and I believe the manuscript is now suitable for publication.